# Effectiveness and Cost-Effectiveness of Case Management in Advanced Heart Failure Patients Attended in Primary Care: A Systematic Review and Meta-Analysis

**DOI:** 10.3390/ijerph192113823

**Published:** 2022-10-24

**Authors:** Caterina Checa, Carlos Canelo-Aybar, Stefanie Suclupe, David Ginesta-López, Anna Berenguera, Xavier Castells, Carlos Brotons, Margarita Posso

**Affiliations:** 1Doctoral Program in Methodology of Biomedical Research, Public Health in Department of Pediatrics, Obstetrics and Gynecology, Preventive Medicine and Public Health, Universitat Autònoma de Barcelona (UAB), 08193 Bellaterra, Spain; 2Fundació Institut Universitari per a la Recerca a l’Atenció Primària de Salut Jordi Gol i Gurina (IDIAPJGol), 08007 Barcelona, Spain; 3Primary Healthcare Centre Dreta de l’Eixample, 08013 Barcelona, Spain; 4Iberoamerican Cochrane Centre, Department of Clinical Epidemiology and Public Health, Biomedical Research Institute Sant Pau (IIB Sant Pau), Sant Antonio María Claret 167, 08025 Barcelona, Spain; 5Department of Clinical Epidemiology and Public Health, de la Santa Creu i Sant Pau (IIB Sant Pau) University Hospital, 08041 Barcelona, Spain; 6Department of Epidemiology and Evaluation, IMIM (Hospital del Mar Medical Research Institute), 08003 Barcelona, Spain; 7Biomedical Research Institute (IBB Sant Pau), Sardenya Primary Health Care Center, 08025 Barcelona, Spain

**Keywords:** case management, advanced heart failure, cost-effectiveness, meta-analyses, mortality, quality of life, hospital admissions, self-care

## Abstract

Aims: Nurse-led case management (CM) may improve quality of life (QoL) for advanced heart failure (HF) patients. No systematic review (SR), however, has summarized its effectiveness/cost-effectiveness. We aimed to evaluate the effect of such programs in primary care settings in advanced HF patients. We examined and summarized evidence on QoL, mortality, hospitalization, self-care, and cost-effectiveness. Methods and results: The MEDLINE, CINAHL, Embase, Clinical Trials, WHO, Registry of International Clinical Trials, and Central Cochrane were searched up to March 2022. The Consensus Health Economic Criteria instrument to assess risk-of-bias in economic evaluations, Cochrane risk-of-bias 2 for clinical trials, and an adaptation of Robins-I for quasi-experimental and cohort studies were employed. Results from nurse-led CM programs did not reduce mortality (RR 0.78, 95% CI 0.53 to 1.15; participants = 1345; studies = 6; I^2^ = 47%). They decreased HF hospitalizations (HR 0.79, 95% CI 0.68 to 0.91; participants = 1989; studies = 8; I^2^ = 0%) and all-cause ones (HR 0.73, 95% CI 0.60 to 0.89; participants = 1012; studies = 5; I^2^ = 36%). QoL improved in medium-term follow-up (SMD 0.18, 95% CI 0.05 to 0.32; participants = 1228; studies = 8; I^2^ = 28%), and self-care was not statistically significant improved (SMD 0.66, 95% CI −0.84 to 2.17; participants = 450; studies = 3; I^2^ = 97%). A wide variety of costs ranging from USD 4975 to EUR 27,538 was observed. The intervention was cost-effective at ≤EUR 60,000/QALY. Conclusions: Nurse-led CM reduces all-cause hospital admissions and HF hospitalizations but not all-cause mortality. QoL improved at medium-term follow-up. Such programs could be cost-effective in high-income countries.

## 1. Introduction

Heart failure (HF) occurs when blood flow is insufficient to meet tissue metabolic needs [1]. Advanced HF (stage D according to America Guidelines) is defined by the presence of symptoms at minimal effort/rest, or hypoperfusion, despite optimal treatment [2,3]. The New York Heart Association (NYHA) also provides a broadly used method of classifying HF severity; advanced HF corresponds to classes III to IV [4]. In 2017, data from the European Heart Failure Registry reported 8.1% and 28.1% for 12-month mortality and hospitalization, respectively, with NYHA III and IV representing a strong predictor of mortality [5].

Advanced HF involves multiple hospital admissions and increased costs for both acute and stable phases [2]. An economic study performed in an HF population in the United States estimated a lifetime cost of USD 126,819 per patient representing around 1–2% of the healthcare budget [6].

Furthermore, quality of life (QoL) is also affected. A systematic review indicated moderate/poor QoL, particularly in elderly and female populations [7], and a greater deterioration in those with advanced HF [8].

One strategy is the nurse-led case management (CM) model, which is a collaborative process of assessment, planning, facilitation, care coordination, evaluation, and advocacy for options and services to meet an individual’s and family’s comprehensive health needs through communication and available resources to promote patient safety, quality of care, and cost-effective outcomes [9,10].

Implemented in community settings, it has achieved improvements in HF-related outcomes and QoL. A 2019 Cochrane systematic review summarized the evidence of all types of CM models for HF patients in all stages. It suggested they be effective in reducing hospitalizations and all-cause mortality [11].

Nevertheless, the effectiveness of CM in advanced HF-related outcomes is controversial. Rogers et al. reported that an interdisciplinary intervention could improve QoL [12] whilst another study reported it did not ameliorate hospitalizations/mortality [13].

To date, no systematic review has summarized the effect of nurse-led CM models on an advanced HF population. We aimed therefore to evaluate the effect of such programs in primary care settings for advanced HF patients and their effect on QoL, mortality, hospitalization, costs, and cost-effectiveness outcomes.

## 2. Methods

### 2.1. Study Inclusion Criteria

#### 2.1.1. Types of Design

Prospective studies with control groups as randomized controlled trials, quasi-experimental, and cohort studies.

#### 2.1.2. Types of Participants

Patients ≥18 years with advanced HF, III/IV NYHA classification, stage D of the American College of Cardiology Foundation/American Heart Association (ACCF/AHA), or under palliative care.

#### 2.1.3. Types of Interventions

Inclusion criteria:Studies where the nurse-led CM model effect was measured.Community interventions including those commencing in hospital.

Exclusion criteria:Nurse-led CM interventions developed only in hospitals.Cardiac rehabilitation programs, unless providing elements of nurse-led CM.Community interventions from specialized HF clinics directed by cardiologists.Only one educational session, without follow-up phone calls/patient interaction.

#### 2.1.4. Type of Comparator/Control

Studies comparing the intervention with usual care or another nurse-led CM program within primary/community care.

#### 2.1.5. Outcomes

##### Primary Outcome

Nurse-led CM program effects on mortality in primary care settings on advanced HF patients.

##### Secondary Outcomes

Results regarding QoL, hospitalization, adherence to treatment, undesirable effects, costs, and cost-effectiveness.

##### Types of Outcome Measures

QoL measured by EuroQol-5D, SF-8, SF-36, and the Kansas City Cardiomyopathy Questionnaire (KCCQ) scales, etc.All-cause and HF mortality.Number of HF hospitalizations or for any other cause during follow-up.Self-care measured by the Appraisal of Self-care Agency (ASA) Scale, European Heart Failure Self-care Behavior Scale, and Self-care of Heart Failure Index.Costs associated with health resources.Cost per QALY (quality-adjusted life year), cost per year of life gained.

Outcomes were measured by follow-up time when available (<6 months, 6–12 months, >12 months), age, and type of consultations (home visits/telemedicine).

### 2.2. Search Methods

#### 2.2.1. Electronic Searches

Searches were performed with MEDLINE, CINAHL, Embase, Clinical Trials, WHO, Registry of International Clinical Trials, and Central Cochrane. The World Health Organization’s International Clinical Trials Registry (ICTRP) platform (http://apps.who.int/trialsearch/, accessed on 24 March 2022), and the ISRCT registry (https://www.isrctn.com/, accessed on 24 March 2022) were used. One hundred ongoing studies were identified by the USA ClinicalTrials.gov registry (https://ClinicalTrials.gov/, accessed on 24 March 2022), however, there were no partial results published and they were excluded.

The database was EndNote χ^2^ software. Publications were included up to March 2022 (See Appendix B).

#### 2.2.2. Other Resources

A manual inspection of the references in previous systematic reviews of HF patient nurse-led CM models was conducted. Gray literature was reviewed, and experts consulted.

### 2.3. Data Collection and Analysis

#### 2.3.1. Study Selection

Initial screening of titles/abstracts was performed by a reviewer. A random sample of 20% of the retrieved references was evaluated by a second reviewer in order to guarantee the quality of the process.

Two reviewers then independently assessed eligibility of the 405 studies based on full-text reading. In case of discrepancy, there were consensus sessions. The Rayyan program [14] was employed throughout.

A PRISMA flow chart depicts the study selection (Figure 1). For excluded studies at the full-text level, see Appendix A.

#### 2.3.2. Data Extraction/Management

Data included author/s, publication year, design, sample and intervention characteristics, and outcomes. Funding information was from economic studies. Data extraction was performed in duplicate.

#### 2.3.3. Risk-of-Bias Assessment

The Cochrane risk-of-bias 2 tool (RoB2) [15] was employed for clinical trials and the Risk-of-Bias in Non-randomized Studies of Interventions (Robins-I) [16] for cohort studies.

The Robins-I was adapted for quasi-experimental studies (Appendix A). Disease decline over time was assumed as a confounder and penalized if the participants were overstable/decompensated during follow-up. In the risk of bias due to intervention classification, intervention/inclusion criteria were either clearly predefined or not.

The Consensus Health Economic Criteria (CHEC) list was employed for economic evaluations [17].

All studies were peer-reviewed, and in the case of any discrepancies, consensus was reached. In studies that measured more than one outcome, these were assessed separately in the outcome domains of the tools.

#### 2.3.4. Intervention Characteristics

Interventions were classified as basic or intensive and determined according to staff availability and issue management during follow-up. Table 1 depicts the intervention and characteristics.

Subgroup analyses were carried out by age ranges (>85 years, 65–85 years, <65 years) and time to follow-up (<6 months, 6–12 months, and >12 months). Population characteristics and outcome measurement instruments were evaluated.

#### 2.3.5. Data Synthesis and Registry

The analysis was an intention-to-treat approach, and all participants were included to reduce the potential selection bias.

Outcome data were evaluated at <6 months, 6–12 months, and >12 months follow-up when available.

Mortality/hospitalization were meta-analyzed using Review Manager (RevMan, version 5.3.5., Cochrane Collaboration, Oxford, UK) and STATA software (v.14.0, STATA Corp, College Station, TX, USA). Pooled relative risk ratios (RRs) and standard mean differences (SMDs) for binary and continuous outcomes were evaluated with the random effect model approach. When means and standard deviations (SDs), or changes of means and SDs from baseline were not reported, they were calculated using standard errors (SE), confidence intervals (CI), or the correlation coefficient.

Magnitude of heterogeneity was assessed using Higgins’s I^2^ statistics and interpreted according to the Cochrane Handbook (0–40%: low, 30–60%: moderate, 50–90%: substantial, 75–100%: considerable). Meta-analysis forest plots for consistency were inspected, given that I^2^ statistics might be artificially inflated when effect estimates from primary studies were very precise [41].

For all meta-analyses with at least 10 included studies, the publication bias was assessed by a visual inspection of Begg’s funnel plot and statistically, using Egger’s test for small study effects (funnel plot asymmetry).

Systematic Reviews of Economic Evaluations guides were followed to analyzed costs and the cost-effectiveness of primary studies [42].

The study was registered in the International Prospective Register of Systematic Reviews (PROSPERO) and published with ID number CRD42020160810.

## 3. Results

### 3.1. Selection of Primary Studies

From 5944 records from four databases, 2129 studies remained. We then reviewed 405 full-text articles: 14 were selected for synthesis. We also reviewed all primary studies obtained from 55 systematic reviews identified in the title/abstract screening and selected 16. In total, 30 studies were included in the evidence synthesis. Of these 30, 25 described the benefits/risks of a nurse-led CM model, and 5 were economic evaluations (Figure 1).

### 3.2. Study Characteristics

#### 3.2.1. Evidence of Effects (Benefits and Risks)

The 25 included studies were published between 1997 and 2016 (Table 1). The majority (17) were from the US and European countries. Most were randomized controlled trials, except for five quasi-experimental studies and one prospective cohort.

Populations were mainly men >60 years. The identification was performed primarily at the hospital or community level. In 22 of the 25 included studies, the comparator was usual care. Most studies had a follow-up of more than six months, with a maximum of one year.

Twelve intensive and thirteen basic programs compared their effect with usual care. The intensity classification was based on the number of contacts made with the patients, staff availability, and to what extent they addressed the issues in the follow-up visits. Telemedicine and home-visit interventions were mainly intensive programs whereas others (clinical consultations, phone calls) were basic.

#### 3.2.2. Cost-Effectiveness Studies

Five economic evaluations were identified. Three were cost-effectiveness studies, and two cost-benefit analyses. All had been performed in high-income countries, four in European ones and another in the US (Table 2).

### 3.3. Quality of Included Studies

#### 3.3.1. Randomized Control Trials

The RoB2 Cochrane [15] tool was used to evaluate the risk of bias (Appendix A). Most studies presented issues with random sequence generation; however, baseline group characteristics did not suggest a randomization concern.

To evaluate the risk of bias of the reported results, all study original protocols were examined to compare the planned statistical analysis with the final result. In nine studies, the protocol was missing and thus referred to as a lack of information with some concerns. Most studies with protocol (8 out of 10) were assessed as a low risk of bias. Hospitalization/mortality was considered a low risk of bias. QoL and self-care outcomes presented some concerns as the interventions were not blinded, and the questionnaires were generally self-reported (Appendix A).

#### 3.3.2. Nonrandomized Trials

The risk of bias in the quasi-experimental and cohort studies was assessed with the ROBINS-I tool (16) (Appendix A). Only one cohort study was identified (Schellinger 2011 [35]). Studies were classified as having a high risk of bias when patients were too stable/decompensated during follow-up. If no control group was present, the progression of advanced HF was considered to play a role in the intervention effectiveness.

Since nonrandomized control trials had lower levels of evidence than randomized ones, they were not included in the meta-analysis, although descriptively reported.

#### 3.3.3. Economic Evaluations

The CHEC tool [17] was used to assess the economic evaluations (Table 2). All studies had a clear research question, with a well-defined population. The economic evaluations were considered as social ones, since they included costs related to patient care beyond hospital admissions. The quality of such studies was therefore downgraded.

Intervention cost-effectiveness was taken to be >6 months although three of the five studies had a shorter follow-up time. Only one study declared its source of funding (Sahlen et al. [47]).

Overall, three economic evaluations had a moderate/low risk of bias, and two were high risk.

### 3.4. Evidence of Effects

#### 3.4.1. All-Cause Mortality

Six studies reported all-cause mortality and indicated no improvement (RR 0.78, 95% CI 0.53 to 1.15; participants = 1345; studies = 6; I^2^ = 47%, low risk of bias).

The follow-up was 12 months in three studies and 6 months in the others. To avoid one death, 32.15 patients were required (Figure 2).

The forest plot did not suggest a marked heterogeneity, nor did the subgroup analysis by length of follow-up indicate differences amongst subgroups (*p* = 0.34). There were, however, some differences in the type of CM (*p* = 0.07). Telemedicine was more effective than home visits (RR 0.47, 95% CI 0.27 to 0.83; participants = 2686 in two studies I^2^ 0%, low risk of bias) with 6 and 12 months of follow-up (Goldberg 2003 [28] and Lynga 2012 [13], respectively) (Appendix A).

#### 3.4.2. Mortality for Heart Failure

None of the studies reported deaths due to HF.

#### 3.4.3. Hospitalizations for Heart Failure

Eight studies described HF hospitalizations and results showed CM as effective in avoiding them (HR 0.79, 95% CI 0.68 to 0.91; participants = 1989; studies = 8; I^2^ = 0%, low risk of bias).

Five studies lacked full information, and two were not randomized controlled trials, and thus excluded. Nevertheless, the results showed that CM was beneficial for advanced HF (Appendix A).

#### 3.4.4. All-Cause Hospitalizations

Five studies reported hospitalizations for all causes and demonstrated CM as protective for this outcome (HR 0.73, 95% CI 0.60 to 0.89; participants = 1012; studies = 5; I^2^ = 36%, low risk of bias).

The subgroup analysis by age, time to follow-up, and CM type did not suggest any differences among groups (*p* = NA, *p* = 0.05 and *p* = 0.40, respectively) (Appendix A).

Seven studies were excluded since four were not randomized controlled trials and three lacked information. Nevertheless, results indicated the benefits of nurse-led CM (Appendix A).

#### 3.4.5. Quality of Life

Eight studies reported QoL and indicated a beneficial effect (SMD 0.18, 95% CI 0.05 to 0.32; participants = 1228; studies = 8; I^2^ = 28%, moderate risk of bias) (Figure 2).

Subgroup differences in follow-up showed that the beneficial effect started at 6 months but was lost at 12 months (*p* = 0.02). In addition, testing for subgroup differences in the type of nurse-led CM suggested an improvement in home visits rather than telemedicine or other means (*p* = 0.02) (Appendix A).

#### 3.4.6. Self-Care

Three studies reported self-care and indicated a statistically nonsignificant beneficial effect. The heterogeneity among studies was high (SMD 0.66, 95% CI −0.84 to 2.17; participants = 450; studies = 3; I^2^ = 97%, moderate risk-of-bias).

### 3.5. Costs and Cost-Effectiveness of Nurse-Led CM

#### 3.5.1. Cost of the Intervention

Except for one study (Sahlen et al. [47]), all reported that investment in a new intervention was greater than in usual care. Cost varied according to intensity, year of implementation, and country. Related costs were mainly linked to healthcare professionals and telemedicine devices in those studies proposing remote data transfer (Table 2).

#### 3.5.2. Cost-Effectiveness (Cost per QALY)

Three studies reported results in the incremental cost-effectiveness ratio (ICER) per QALY of the intervention compared with usual care. They also presented the lowest risk of bias and the largest time horizon (Postmus et al., Grustam et al. and Sahlen et al. [45,46,47]). They reported that intensive interventions, compared with basic ones/usual care, obtained larger benefits in terms of QALY and LY in NYHA III/IV patients. Regarding the ICER per QALY, ICERs of EUR 59,289 and EUR 14,027, respectively, were observed when comparing intensive versus usual care. Figures were below EUR 60,000/QALY.

#### 3.5.3. Cost-Benefit Studies

Studies reported savings due to fewer hospital admissions. The net benefit was mainly determined by the price of the intervention.

## 4. Discussion

This systematic review summarized the quality of evidence regarding the effectiveness/cost-effectiveness of nurse-led CM programs in advanced HF populations. We included 30 studies, 25 reported effectiveness (19 randomized controlled trials, 5 quasi-experimental, and 1 cohort), and 5 economic evaluations. Only meta-analyzed studies with a low risk of bias, or with the lowest risk of bias available, were analyzed. Nonrandomized trials or studies lacking data were excluded. The latter were presented as narrative results and showed the same direction of effectiveness.

Our results were nonsignificant to indicate that nurse-led CM intervention reduced all-cause mortality. Interventions with telemedicine were the most effective. No study reported mortality due to HF.

Regarding HF hospitalizations, we found eight low-risk-of-bias randomized controlled trials. Five additional studies were narratively summarized. Nurse-led CM, telemedicine, and home visits were effective in preventing HF and all-cause hospitalizations.

Eight studies reported an improvement in QoL at 6 months which did not extend at 12 months. Three studies suggested that patients in the program had better self-care, although this was not statistically significant and the intervention costs among studies ranged from USD 4975 (2003-year value) to EUR 27,538 (2015-year value).

The most recent similar review (Takeda) explored different CM interventions for all-stage HF patients. Whilst our results concurred, they were not statistically significant, in contrast to other authors [11,48,49]; the fact that our population was at the final stage of the disease may have played a role.

In a similar manner to Bashi et al. [50], we found that patients with lower mortality were those who received telemedicine. Such results are, however, controversial as Flodgren et al. described no differences with usual care [51]. Reasons for this may include sociocultural differences, and in this sense, further research is required.

We observed that the nurse-led CM interventions reduced the risk of hospitalizations. This is a relevant finding since hospitalizations for advanced HF are common [52] and avoiding hospitalization can also reduce mortality [53].

In agreement with Rice et al. [54], QoL improved with the intervention. We found, however, that in our population this was at 6 months after the intervention and only lasted up to 12 months. Nevertheless, due to the advanced stage of the disease, we believe any gain, or even maintenance, in the QoL of patients with advanced HF to be relevant.

QoL did not improve in the telemedicine group, concurring with Bauce et al. [55]. Personal contact with healthcare professionals can produce a certain emotional proximity which may have a positive impact on QoL and should be further evaluated.

Nurse-led CM interventions could also improve self-care [56]. Nevertheless, our findings were not statistically significant, and there was considerable heterogeneity among studies. The intervention effect was lost with time as patients lost motivation. Factors favoring long-term self-care should be further explored as they have an impact on the reduction of hospital admissions [57].

Nurse-led CM could be cost-effective, a finding that concurs with Rice et al. [54], probably due to the savings from fewer hospitalizations. In terms of QALYs, Fergenbaum et al. concluded that a home-based intervention improved the QALY by 0.11 and reduced costs [58]. In our review, all the studies that reported QALYs described improvements above the figure described by Fergenbaum except for the Postmus study.

Advanced HF patients require more resources to improve their QALY thus increasing incremental cost. Nurse-led CM was not found to be particularly cost-effective, nevertheless, a threshold of EUR 60,000/QALY may be considered affordable for high-income countries.

Further studies should consider differentiating advanced HF from the general HF population, since this subgroup has different needs.

### Limitations

The limitations of this systematic review are mostly derived from those of the primary included studies. We found seven, five, and four studies corresponding to all-cause hospitalizations, HF hospitalizations, and QoL, respectively, with concerns of a high risk of bias leading to their exclusion from the pooled analysis. We did, however, narratively summarize these data and found similar results in most cases.Nurse-led CM interventions may have varying characteristics according to their settings which could result in heterogeneity. For clarification, we created a descriptive table with all the characteristics of each intervention.The CM overall effect can be affected over time. We observed a short-term beneficial effect that was depleted on the medium/long term. We therefore carried out the meta-analysis with different follow-up time groups to analyze this factor.

## 5. Conclusions

Nurse-led CM can reduce all-cause hospital admissions and HF hospitalizations but not all-cause mortality. QoL improved in medium-term follow-up, and better self-care/survival was reported, although it was not statistically significant. The intervention could be cost-effective for less than EUR 60,000/QALY. More intensive nurse-led case management studies are needed to determine the cost-effectiveness of the program.

## Figures and Tables

**Figure 1 ijerph-19-13823-f001:**
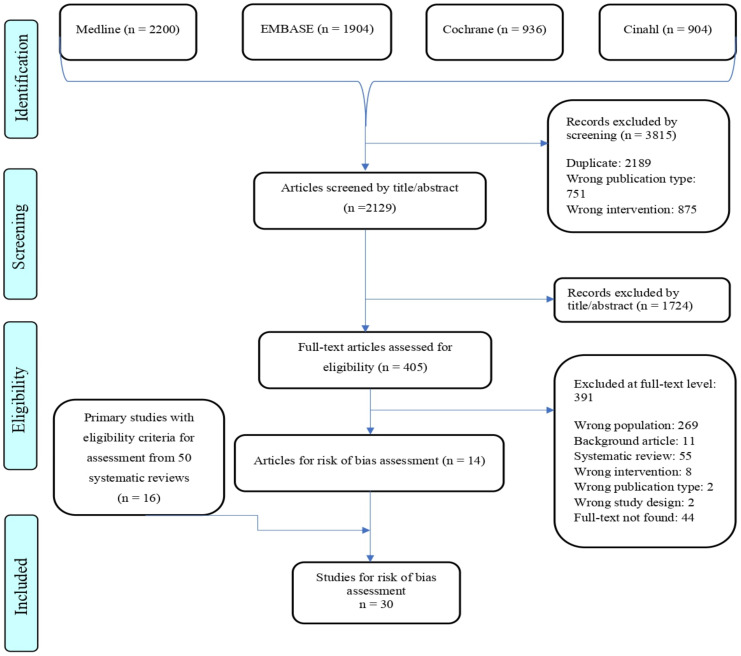
Study flow chart of studies evaluating clinical efficacy.

**Figure 2 ijerph-19-13823-f002:**
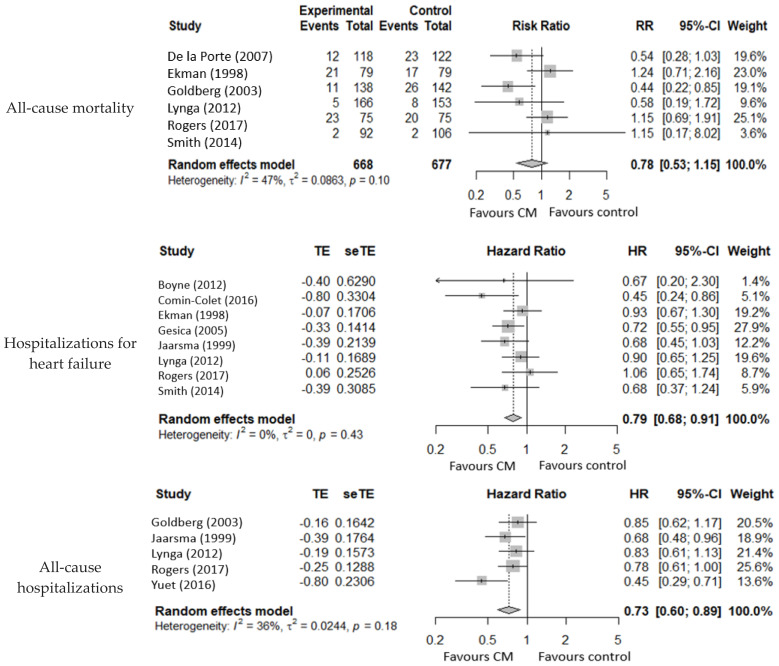
Metanalysis for all-cause mortality, hospitalizations for heart failure, all-cause hospitalizations, quality of life, and self-care. CM: nurse-led case management.

**Table 1 ijerph-19-13823-t001:** Characteristics and summary of results of the included studies for the evaluation of case management in advanced heart failure patients.

		Studies Evaluating Clinical Efficacy						
Author, Year Country	Study Design (Number of Subjects Included)	Mean Age (Standard Deviation) Case Management vs. Control	Gender (% Women) Case Management vs. Control	Population	Intervention Characteristics and Main Component *	Control	Maximum Follow-Up Time (Days)	Outcome Measures
Aiken, 2006 [18] USA	RCT (N = 129)	68 (14) vs. 70 (13) ^β^	58 vs. 70	Patients from community or hospitalized with chronic heart failure in NYHA III or IV	Intensive intervention ^b,f^ Home visits	Usual care	270	Quality of life
Bondmass, 2007 [19] USA	RCT (N = 186)	62.1 (13.9) vs. 62.8 (12.4)	63.3 vs. 60.4	Patients hospitalized with HF in NYHA III or IV	Intensive intervention ^c,f^ Telemedicine	NHV: Nurse home visits	90	Treatment adherence, quality of life
Boyne, 2012 [20] Netherlands	RCT Total: (N = 382) Subgroup NYHA III: 153 Subgroup NYHA IV: 10	Total: 71.0 (11.9) vs. 71.9 (10.5) Subgroup in NYHA III and IV: not reported	Total: 42 vs. 40 Subgroup in NYHA III and IV: not reported	Patients in the community diagnosed with HF >18 years and being treated by an HF nurse and a cardiologist in an HF clinic Subgroup analysis in NYHA III and IV	Intensive intervention ^a,c,e^ Telemedicine	Usual Care	365	Hospitalizations for HF
Brännström, 2014 [21] Sweden	RCT (N = 72)	81.9 (7.2) vs. 76.6 (10.2)	27.8 vs. 30.6	Patients in the community diagnosed with HF in NYHA III or IV	Intensive intervention ^b,f^ Home visits	Usual Care	180	All-cause hospitalizations, quality of life, self-efficacy
Comin-Colet, 2016 [22] Spain	RCT (N = 178)	Total: 74 (11) vs. 75 (11) Subgroup in NYHA III and IV: not reported	Total: 43 vs. 39 Subgroup in NYHA III and IV: not reported	Patients hospitalized with HF Subgroup analysis in NYHA III and IV	Intensive intervention ^c,d,f^ Telemedicine	HF program	180	Hospitalizations for HF
De la Porte, 2007 [23] Netherlands	RCT (N = 340)	70 (10) vs. 71 (10)	34 vs. 21	Patients in the community or hospitalized with NYHA III or IV	Intensive intervention ^a,f,g^ Clinical consultations	Usual Care	365	All-cause mortality, hospitalizations for HF, self-care, and quality of life
Delaney, 2010 [24] USA	Quasi-experimental studio with control without randomization (N = 24)	Overall: 79.04 (11.8) ^π^	58.3 vs. 58.3	Patients with a primary diagnosis of HF in NYHA III or IV	Intensive intervention ^b,c^ Telemedicine	Usual care	90	Hospitalizations for HF, quality of life
Ekman, 1998 [25] Sweden	RCT (N = 158)	Overall: 80.3 (6.8) ^π^	42 ^π^	Patients hospitalized with HF in NYHA III or IV	Basic intervention ^a,f,g^ (office hours) Clinical consultations	Usual care	180	All-cause mortality, hospitalizations for HF, all-cause hospitalizations
Fonarow, 1997 [26] USA	Quasi-experimental pre-post (N = 214)	52.6 (10)	19	Patients in the community diagnosed with HF in NYHA III or IV and potential candidates for transplantation	Basic intervention ^a,f^ Clinical consultations	Usual care	180 pre and 180 post	All-cause mortality, hospitalizations for HF
GESICA, 2005 [27] Argentina	RCT Total: (N = 1518) Subgroup NYHA III or IV: (N = 750)	Total: 64.8 (13.9) vs. 65.2 (12.7) Subgroup: not reported	Total: 27.4 vs. 31.1 Subgroup in NYHA III and IV: not reported	Patients in the community diagnosed with HF and >18 years Subgroup analysis in NYHA III and IV	Basic intervention ^f^ Phone calls	Usual Care	From 180 to 365	Hospitalizations for HF
Goldberg, 2003 [28] USA	RCT (N = 280)	57.9 (15.7) vs. 60.2 (14.9)	30.4 vs. 34.5	Patients hospitalized with HF in NYHA III or IV	Intensive intervention ^c^ Telemedicine	Usual Care	180	All-cause mortality, hospitalizations for HF, all cause hospitalizations and quality of life
Holst, 2001 [29] Australia	Quasi-experimental (N = 42)	54 (13)	16.6	Patients with NYHA III or IV	Basic intervention ^a^ Clinic consultations	Usual care	180	All-cause hospitalizations, quality of life
Jaarsma, 1999 [30] Netherlands	RCT (N = 179)	73 (9) vs. 73 (9)	44 vs. 41	Patients hospitalized for HF with NYHA III or IV	Basic intervention ^b,g^ Home visits	Usual care	270	Hospitalizations for HF, All-cause hospitalizations, treatment adherence
Jaarsma, 2000 [31] Netherlands	RCT (N = 132)	72 (9) vs. 72 (10)	45 vs. 36	Patients admitted in cardiology unit for HF with NYHA III or IV	Basic intervention ^a,b,f,g^ Home visits	Usual care	270	Self-care, quality of life
Lynga, 2012 [13] Sweden	RCT (N = 319)	73.7 (9.9) vs. 73.5 (10.4)	24.1 vs. 26.1	Patients hospitalized for HF with NYHA III or IV	Basic intervention ^a,c,g^ Telemedicine	Usual Care	Up to cardiac hospitalization or 365 days	All-cause mortality, hospitalizations for HF, all-cause hospitalizations
Man, 2018 [32] China	RCT (N = 84)	78.3 (16.8) vs. 78.4 (10)	56.1 vs. 39	Patients hospitalized for HF with NYHA III or IV	Intensive intervention ^b,f^ Home visits	Usual care	90	Quality of life
McDonald, 2001 [33] Ireland	RCT (N = 70)	69.9 (11.3) vs. 67.9 (12)	14.3 vs. 18.6	Patients hospitalized with HF and NYHA III or IV	Basic intervention ^a,f,g^ Clinical consultations	Usual care	30	Hospitalizations for HF, all-cause mortality
Ong, 2016 [34] USA	RCT (N = 1437)	Median (interquartile range) Total: 73 (62–84) vs. 74 (63–82) Subgroup in NYHA III and IV: not reported	Total: 50.2 vs. 50.5 Subgroup in NYHA III and IV: not reported	Patients admitted to hospital for decompensated HF and >50 years old Subgroup analysis in NYHA III and IV	Intensive intervention ^c,f,g^ Telemedicine	Usual care	180	All-cause hospitalizations
Rogers, 2017 [12] USA	RCT (N = 150)	71.9 (12.4) vs. 69.8 (13.4)	44 vs. 50.7	Patients hospitalized for HF or within 2 weeks of discharge and dyspnea at rest or minimal exertion	Intensive intervention ^b^ Not clearly reported	Usual Care	180	All-cause mortality, hospitalizations for HF, all-cause hospitalizations, quality of life
Schellinger, 2011 [35] USA	Cohort study (N = 1894)	75.63 vs. 73.84 ^∞^	52 vs. 48.4 ^∞^	Patients with a primary or secondary HF diagnosis in community setting	Basic intervention ^a^ Clinical consultations	Usual care	60	All-cause hospitalizations
Shah, 1998 [36] USA	Quasi-experimental (N = 27) Subgroup NYHA III and IV (N = 17)	62 (range 42–81)	0	Patients hospitalized for HF	Basic intervention ^f,g^ Phone calls	Usual care	365	All-cause hospitalizations
Smith, 2014 [37] USA	RCT (N = 198)	62.6 (14.1) vs. 62.1 (12.5)	44 vs. 34	Patients hospitalized with HF in NYHA III or IV	Basic intervention ^a^ Clinical consultations	Usual Care	365	All-cause mortality, hospitalizations for HF
Vavouranakis, 2003 [38] Greece	Quasi-experimental (N = 33)	65.4 (6.7)	12.1	Patients in the community with HF and NYHA III or IV	Basic intervention ^b,f,g^ Home visits	Usual care	365	All-cause hospitalizations, quality of life
Yuet, 2016 [39] China	RCT (N = 84)	78.3 (16.8) vs. 78.4 (10.0)	55.1 vs. 39	Patients hospitalized with HF in NYHA III or IV	Intensive intervention ^b,f^ Home visits	Two placebo calls from assistant unrelated to clinical issues	90	All-cause hospitalizations, quality of life
Zamanzadeh, 2013 [40] Iran	RCT (N = 78)	65.82 (9.87) vs. 61.63 (12.47)	42.1 vs. 52.5	Patients diagnosed with HF in NYHA III or IV and an ejection fraction <40%	Basic intervention ^a,f,g^ Clinical consultations	Usual Care	90	Self-care (treatment adherence)

RCT: randomized control trial. HF: heart failure. NYHA: New York Heart Association. * This classification was based on the number of contacts made with the patients, staff availability, and the extent to which they addressed the issues during follow-up. The main component of each intervention was also described: ^a^—clinical consultations, ^b^—home visits, ^c^—remote vital sign monitoring, ^d^—videophone, ^e^—messaging, ^f^—scheduled telephone calls, ^g^—telephone availability of staff (unscheduled). ^∞^ There is no exact information about the “no program” group but it seems to be similar to the uncompleted follow-up. ^π^ Case management vs. control not reported. ^β^ Population with advanced chronic diseases. Advanced HF disease not reported.

**Table 2 ijerph-19-13823-t002:** Characteristics and summary of results of the included studies for the economic evaluation of case management in advanced heart failure patients.

Studies Evaluating the Economic Evidence for Nurse Case Management											
Author, Year	Study Design, Country	Population	Intervention Characteristics *	Control	Time Horizon, Perspective	Difference in Cost (Year Value)	Difference in Outcome		ICER		Risk of Bias (CHEC) Score)
Gregory, 2006 [43]	Cost-benefit, USA	Patients admitted to hospital with primary diagnosis of HF in NYHA III or IV	Intensive intervention: 2, 6, 7 (24 h availability)	SC	90 days, healthcare perspective	Reference SC: USD 3979 (2003) USA Dollars Intensive vs. SC: USD 996 additional cost per patient	USD −759 due to reduction in hospitalization costs per patient		USD +237 not considered cost saving		11.5/19
Ledwidge, 2003 [44]	Cost-benefit, Ireland	Patients admitted to hospital with a diagnosis of HF in NYHA IV	Basic intervention: 1, 6, 7 (working hours)	SC	3 months, healthcare perspective	Reference SC: no cost Basic vs. SC: EUR 113 (1999) additional cost per patient	EUR −43,955 due to reduction in hospitalization costs per patient		Net saving EUR −379.75 per patient		12/19
Postmus, 2011 [45]	Cost-effectiveness, Sweden	Patients admitted to hospital with primary diagnosis of HF in NYHA III or IV	Basic intervention: 1, 7 (working hours) Intensive intervention: 1, 2, 6, 7 (24 h availability)	SC	18 months, healthcare perspective	Reference SC: EUR 10,692 per patient (2009)	QALY	LY	QALY (cost/QALY)	LY (cost/LY)	14/19
						Basic vs. SC: EUR 1101 additional cost per patient (2009)	Basic vs. SC: 0.014	Basic vs. SC: 0.042	Basic vs. SC: EUR 77,335	Basic vs. SC: EUR 25,923	
						Intensive vs: SC: EUR 1770 additional cost per patient (2009)	Intensive vs. SC: 0.029	Intensive vs. SC: 0.057	Intensive vs. SC: EUR 59,289	Intensive vs. SC: EUR 30,933	
						Intensive vs. basic: EUR 669 additional cost per patient (2009)	Intensive vs. basic: 0.015	Intensive vs. basic: 0.014	Intensive vs. basic: EUR 42,839	Intensive vs. basic: EUR 45,219	
Grustam, 2018 [46]	Cost-effectiveness Markov model, Netherlands	Patients > 70 years admitted to hospital with a diagnosis of HF in NYHA IV	Basic intervention (nurse telephone support): 1, 6, 7 (working hours) Intensive intervention (home telemonitoring): 3	SC	Lifetime (20 years), health system perspective	Reference SC: EUR 15,407 per patient (2015)	QALY	LY	QALY (cost/QALY)	LY (cost/LY)	16/19
						Basic vs. SC: EUR 7042 additional cost per patient (2015)	Basic vs. SC: 0.75	Basic vs. SC: 0.96	Basic vs. SC: EUR 9398	Basic vs. SC: EUR 7364	
						Intensive vs. SC: EUR 12,131 additional cost per patient (2015)	Intensive vs. SC: 0.86	Intensive vs. SC: 1.14	Intensive vs. SC: EUR 14,027	Intensive vs. SC: EUR 10,644	
						Intensive vs. basic: EUR 5090 additional cost per patient (2015)	Intensive vs. basic: 0.12	Intensive vs. basic: 0.18	Intensive vs. basic: EUR 44,040	Intensive vs. basic: EUR 27,733	
Sahlen, 2016 [47]	Cost-effectiveness, Sweden	Patients diagnosed with HF in NYHA III or IV and attended in the community	Intensive intervention: 2, 6	SC	6 months, healthcare perspective	Reference SC: EUR 5727 per patient (2012) Intensive vs. SC: EUR −1649 saving cost per patient	0.25 QALY		Dominant		13/19

* Intervention characteristics; ^1^—clinical consultations, ^2^—home visits, ^3^—remote vital signs monitoring, ^4^—videophone, ^5^—messaging, ^6^—scheduled telephone calls, ^7^—telephone availability of staff (unscheduled). ICER: incremental cost-effectiveness ratio. QALY: quality-adjusted life years. LY: life year. CHEC: consensus on health economics criteria checklist. SC: standard care. HF: heart failure. NYHA: New York Heart Association.

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
