# Peer review of "Effectiveness and Cost-Effectiveness of Case Management in Advanced Heart Failure Patients Attended in Primary Care: A Systematic Review and Meta-Analysis"

_ijerph, 2022, doi:10.3390/ijerph192113823_

Round 1

Reviewer 1 Report

Checa and colleagues set out the difficult task to examine effectiveness and cost-effectiveness of case management in advanced heart failure patients attended in primary care. The paper reads well however there are a number of minor typographical, grammatical and styling errors throughout which need addressed. 

I have made a number of points below which also should be addressed prior to publication in no particular order:

1.          In the abstract authors state “Nurse-led CM programs improved mortality although non-statistically significant” and “self-care was non-statistically significant improved”. Similar statements are made in other sections (e.g. conclusions of the abstract, results). These statements are incorrect and presented data do not support them. 

2.          The authors do not clearly define “advanced heart failure”. They refer to advanced HF, ACCF/AHA guidelines (stage D heart failure), NYHA class (Class III/IV) as well as patients under palliative care. These however should not be used interchangeably.

3.          The primary outcome is not specific and the results section only refers to secondary outcomes. 

4.          Line 106- please expand on etc (list other tools for self care measurement)

5.          Authors state there was no previous systematic review to examine the effect of CM however through the manuscript they do refer to other systematic reviews on this topic.

6.          The authors state ongoing studies were identified through relevant resources, however these were excluded as per their inclusion criteria.

7.          Please expand on role of second reviewer- what was the purpose of review of random 20% of titles/abstracts. 

8.          Figure 1 does not display properly- please correct.

9.          The Cochrane risk-of-bias 2 tool is inconsistently referred to through the methods and results- please ensure the acronym and/or full name used is consistent.

10.       In Table 1, column “intervention characteristics and main components”- please re-consider use of superscript to denote components of intervention as it appears to be similar to the referencing style used in the manuscript.

11.       Please expand how 2129 studies were narrowed down to 405 full text articles and 14 which were included in final synthesis. (I understand this may be clear from the Figure 1 however this does not project properly).

12.       Line 200- “In all cases, the comparator was usual care”- this does not appear to be correct statement as some studies in Table 1 employed HF program or telephone consultation.

13.       Line 200/201: In table 1 only 11 studies had a follow up of => 6 months (unless 180 days means 6 months- please clarify and correct if appropriate).

14.       Line 202 please remove duplication. Please also expand on what is considered as “basic” and “intensive” intervention.

15.       Please elaborate how many protocols were examined and how many were missing.

16.       The authors make statements regarding “non statistically beneficial effect” throughout the manuscript. These statements are incorrect i.e. data do not support the benefit if there is no significant difference.

17.       In result section the authors use hospitalisations for HF interchangeably with HF readmissions (and later similar statements on all cause hospitalisation). Please consider using consistent terminology as not every hospitalisation is readmission (some studies included patients in the community, we cannot assume all were previously hospitalised patients).

18.       Please avoid making general statements regarding CM effectiveness on avoidance of admission or protection against mortality (correlation doesn’t always mean causation).

19.       The authors state beneficial effect on QoL was lost after 12 months none of the studies followed up patients for longer than 12 months. 

20.       Figure 2 does not display properly (part of the figure is outside page margin). 

21.       Please consider of meta-analysis of 1 or 2 studies, divided into subgroups with very small numbers, is truly adding value to the manuscript. 

22.       The authors conclude “8 studies reported improvement in QoL at 6 months which did not extend beyond 12 months”. None of these studies followed up patients beyond 12 months. 

23.       Line 320: “four studies suggested…”. Only three studies reported self-care.

24.       The manuscript uses multiple different referencing styles- please keep consistent.

Author Response

1. In the abstract authors state “Nurse-led CM programs improved mortality although non-statistically significant” and “self-care was non-statistically significant improved”. Similar statements are made in other sections (e.g. conclusions of the abstract, results). These statements are incorrect and presented data do not support them. 

Thanks for your comment, we have modified all the sentences in the abstract, results and discussion sections to give a much more cautious message because the RR estimator was 0.78, 95% but CI was 0.53 to 1.15.

2. The authors do not clearly define “advanced heart failure”. They refer to advanced HF, ACCF/AHA guidelines (stage D heart failure), NYHA class (Class III/IV) as well as patients under palliative care. These however should not be used interchangeably.

Thank you for your comment.

We defined advanced HF in the introduction as the stage of the disease with symptoms at minimal effort or at rest despite optimal treatment.

Due to the fact that patients with heart failure are classified according to the severity of the symptoms, we chose those studies where the patients were classified with maximum severity (symptoms at minimum effort or rest) or, in palliative treatment (as long as the HF was in the terminal phase of the disease and they have symptoms at minimal effort or rest).

  1. The primary outcome is not specific and the results section only refers to secondary outcomes. 

Thank you for your comment, we have been reviewing all the outcomes and have modified the primary outcome.

  1. Line 106- please expand on etc (list other tools for self care measurement)

We have specified all the scales in this line.

  1. Authors state there was no previous systematic review to examine the effect of CM however through the manuscript they do refer to other systematic reviews on this topic.

Thank you for your comment. We have found no systematic reviews on the effect of nurse-led case management in the community on patients with advanced HF. However, there are reviews that analyze the effect in the population with HF in all stages. We have clarified this in the introduction for further understanding.

  1. The authors state ongoing studies were identified through relevant resources, however these were excluded as per their inclusion criteria.

Since there were no partial results published, they were excluded. We clarified this in methods section.

  1. Please expand on role of second reviewer- what was the purpose of review of random 20% of titles/abstracts. 

In order to guarantee the quality of the process a random sample of de 20% was also evaluated by a second reviewer. We clarified this in the methods section.

  1. Figure 1 does not display properly- please correct.

We improved figure 1 visualization.

  1. The Cochrane risk-of-bias 2 tool is inconsistently referred to through the methods and results- please ensure the acronym and/or full name used is consistent.

Thank you, we have corrected the error in acronym name in results section.

  1. In Table 1, column “intervention characteristics and main components”- please re-consider use of superscript to denote components of intervention as it appears to be similar to the referencing style used in the manuscript.

Thank you, we changed it and we used letters to as superscript.

  1. Please expand how 2129 studies were narrowed down to 405 full text articles and 14 which were included in final synthesis. (I understand this may be clear from the Figure 1 however this does not project properly).

We improved figure 1 visualization.

  1. Line 200- “In all cases, the comparator was usual care”- this does not appear to be correct statement as some studies in Table 1 employed HF program or telephone consultation.

We rewrote the statement taking into account the number of studies with usual care as a comparator.

  1. Line 200/201: In table 1 only 11 studies had a follow up of => 6 months (unless 180 days means 6 months- please clarify and correct if appropriate).

Yes, in that case, 6 months were 180 follow up days.

  1. Line 202 please remove duplication. Please also expand on what is considered as “basic” and “intensive” intervention.

We removed duplication and we have explained what was considered basic and intensive in the same paragraph.

  1. Please elaborate how many protocols were examined and how many were missing.

We added numbers in this paragraph.

  1. The authors make statements regarding “non statistically beneficial effect” throughout the manuscript. These statements are incorrect i.e. data do not support the benefit if there is no significant difference.

Thanks for your comment, we have modified all the sentences in the abstract, results and discussion sections to give a much more cautious message that data supports.

  1. In result section the authors use hospitalisations for HF interchangeably with HF readmissions (and later similar statements on all cause hospitalisation). Please consider using consistent terminology as not every hospitalisation is readmission (some studies included patients in the community, we cannot assume all were previously hospitalised patients).

Thank you for your comment. We correct the word readmission and we are consistent with hospitalization terminology.

  1. Please avoid making general statements regarding CM effectiveness on avoidance of admission or protection against mortality (correlation doesn’t always mean causation).

Thank you for your comment, in this sense, we are using the same terminology as the primary studies that we have included. Since all the meta-analyses have been carried out with clinical trials, we have used the terminology of causality.

  1. The authors state beneficial effect on QoL was lost after 12 months none of the studies followed up patients for longer than 12 months. 

Thank you for your comment, we have clarified through the manuscript that the effect was lost at 12 months of follow-up.

  1. Figure 2 does not display properly (part of the figure is outside page margin). 

Thank you, we corrected margins.

  1. Please consider of meta-analysis of 1 or 2 studies, divided into subgroups with very small numbers, is truly adding value to the manuscript. 

We have been reviewing all the meta-analyses and we have observed that small studies are in line with larger ones, for this reason we have thought that it would not provide extra information. In any case, if you consider it to be of special relevance, we could carry out a subgroup, however, we would need a little more time to be able to carry out the meta-analyses.

  1. The authors conclude “8 studies reported improvement in QoL at 6 months which did not extend beyond 12 months”. None of these studies followed up patients beyond 12 months. 

Thank you, as mentioned above, we modified all sentences related to this.

  1. Line 320: “four studies suggested…”. Only three studies reported self-care.

We corrected the sentence.

  1. The manuscript uses multiple different referencing styles- please keep consistent.

Thank you, we have corrected some errors in the reference section, please let us know if you detected any other errors.

Reviewer 2 Report

In the manuscript Authors raised an important issue of case management care of advanced heart failure in primary care. They prepared systematic review of and subsequently meta-analysis of nurse-led case management programs and evaluated their effects on QALY, mortality, hospitalization, self-care, and cost-effectiveness. The paper is well-written, study design is appropriate, however there are some major issues that need to be addressed. 

COMMENTS:

MAJOR:

Some of the studies included to evaluation do not meet the inclusion criteria as they involve patients other than NYHA III-IV, Stage D according to ACCF/AHA or under palliative care (advanced heart failure population) and therefore presented results are not reliable, e.g:

·         Ong et al. 2016 enrolled 1437 patients with decompensated HF, among which only 880 pts were NYHA III and IV (this should be removed from metaanalysis);

·         GESICA Investigators. 2005 enrolled 1518 patients with decompensated HF, among which only 750 pts were NYHA III and IV (this should be removed from metaanalysis);

·         Boyne at al. 2012; only 163 (from 382) were NYHA III/IV patients (this should be removed from metaanalysis);

·         Schellinger et al. 2011; there is no information about NYHA group among the study population (this should be removed from metaanalysis);

·         Shah et al enrolled patients with DCM NYHA II-IV (this should be removed from metaanalysis);

Data should be re-analysed after removing abovementioned studies.

MINOR:

ABSTRACT Line 34 number of participants should be included

I2 or I2  – this should be unified in all txt

If mortality did not improved significantly Authors should clearly state that there was no effect on mortality.

Line 96 primary outcome should be clearly stated “effects” is not clear

Figure 1 should be reedited as in this form part of it is missing and it it unreadable

Table 1 Information should be more structured as in this form its hard to read table (as there is no key – no alphabetical order, year)

Why authors decided to mix RCT with “quasi-experimental” studies?

Author Response

Some of the studies included to evaluation do not meet the inclusion criteria as they involve patients other than NYHA III-IV, Stage D according to ACCF/AHA or under palliative care (advanced heart failure population) and therefore presented results are not reliable, e.g:

  • Ong et al. 2016 enrolled 1437 patients with decompensated HF, among which only 880 pts were NYHA III and IV (this should be removed from metaanalysis); subgroup table IV
  • GESICA Investigators. 2005 enrolled 1518 patients with decompensated HF, among which only 750 pts were NYHA III and IV (this should be removed from metaanalysis);
  • Boyne at al. 2012; only 163 (from 382) were NYHA III/IV patients (this should be removed from metaanalysis);
  • Schellinger et al. 2011; there is no information about NYHA group among the study population (this should be removed from metaanalysis);
  • Shah et al enrolled patients with DCM NYHA II-IV (this should be removed from metaanalysis);

Data should be re-analysed after removing abovementioned studies.

Dear reviewer,

Thank you for your comment. As you mentioned, the sample of these studies were people with HF in all phases, nevertheless, in all of them a subgroup analysis has been carried out for patients in NYHA III and IV, and only these results from the subgroup analysis were meta-analyzed.  

MINOR:

  1. ABSTRACT Line 34 number of participants should be included

We included number of participants

  1. I2 or I– this should be unified in all txt

Thank you, we unified the format in all  text.

  1. If mortality did not improved significantly Authors should clearly state that there was no effect on mortality.

Thanks for your comment, we have modified all the sentences in the abstract, results and discussion sections to give a much more cautious message because the RR estimator was 0.78, 95% but CI was 0.53 to 1.15.

  1. Line 96 primary outcome should be clearly stated “effects” is not clear

Thank you for your comment, we have been reviewing all the outcomes and have modified the primary outcome.

  1. Figure 1 should be reedited as in this form part of it is missing and it it unreadabl

We improved the visualization of Figure 1

  1. Table 1 Information should be more structured as in this form its hard to read table (as there is no key – no alphabetical order, year)

Thank you for your recommendation, we have arranged the entire table in alphabetical order.

  1. Why authors decided to mix RCT with “quasi-experimental” studies?

We collected both RCTs and quasi-experimental for our review, however the studies are not mixed as only RCTs were meta-analyzed.

Reviewer 3 Report

This review evaluated the effect of nurse-led case management in primary care settings in advanced HF patients by summarizing evidence on QoL, mortality, hospitalization, self-care, and cost-effectiveness. However, there are a few comments to improve the paper:

1. Avoid abbreviations (example, CHEC) in the abstract.

2. The authors should explain nurse-led case management in the introduction.

3. Figure 1 and all forest plots were not clear.

4. Authors should explain how risk-of-bias assessments were evaluated and why this assessment was separated for each outcome, as shown in the supplemental table S3 and table S4?

5. What are the authors mean by “Heterogeneity was classified as basic or intensive and determined according to staff availability and issue management during follow-up”?  Heterogeneity assessment should be referred to as heterogeneity in the meta-analysis that measured the variation in the study outcome between studies by Cochran’s Q.

6. Only 15 studies over 30 included studies were reported in the meta-analysis (based on the forest plots). How about the rest?

7. How the authors concluded “low risk of bias” in line 247, line 253-254, line 260-261, line 267-268 and “moderate risk of bias” in line 277 and line 285?

8. I recommend having standardized currency so that we can see a clearer differential in the cost.

9. I suggest the authors add meta-analyses for the cost of the interventions, cost-effectiveness and cost benefits.

10. The statement “Results indicated that nurse-led CM interventions improved all-cause mortality without being statistically significant” (line 313-314) was confusing. Why did the authors state the nurse-led CM interventions improved all-cause mortality but were statistically insignificant, which means no difference between nurse-led CM intervention and the control intervention?

11. Authors should include the impact of the small size of the study in the limitation.

12. Authors should add recommendations in conclusion.

Author Response

  1. Avoid abbreviations (example, CHEC) in the abstract.

Thank you, we improve that in the abstract

  1. The authors should explain nurse-led case management in the introduction.

We added an explanation of nurse-led case management in the introduction section.

  1. Figure 1 and all forest plots were not clear.

We improved visualization.

  1. Authors should explain how risk-of-bias assessments were evaluated and why this assessment was separated for each outcome, as shown in the supplemental table S3 and table S4?

Thank you. We added how risk-of-bias was evaluated in methods section.

We have carried out a separate assessment of the outcomes as the tool guide recommends in order to know the specific bias of each outcome. We include the bibliography of the tools below.

Sterne JAC, Savović J, Page MJ, Elbers RG, Blencowe NS, Boutron I et al. RoB 2: a revised tool for assessing risk of bias in randomised trials. BMJ. 2019; 366: l4898.

Sterne JA, Hernán MA, Reeves BC, Savović J, Berkman ND, Viswanathan M, et al. ROBINS-I: A tool for assessing risk of bias in non-randomised studies of interventions. BMJ. 2016;355.

  1. What are the authors mean by “Heterogeneity was classified as basic or intensive and determined according to staff availability and issue management during follow-up”?  Heterogeneity assessment should be referred to as heterogeneity in the meta-analysis that measured the variation in the study outcome between studies by Cochran’s Q.

Thank you for your comment. In that case, we referred to “intervention characteristics”, it was a writing error that we have modified.

  1. Only 15 studies over 30 included studies were reported in the meta-analysis (based on the forest plots). How about the rest?

Of the 30 studies included in the review, 15 were meta-analyzed since they were the studies with the highest methodological rigor. The rest of the studies were described descriptively in supplementary material and were also described narratively in the results section.

Regarding the 5 cost-effectiveness studies, we answer below in question number 9.

  1. How the authors concluded “low risk of bias” in line 247, line 253-254, line 260-261, line 267-268 and “moderate risk of bias” in line 277 and line 285?

This conclusion is related to the results of the risk of bias assessment in each outcome of the primary studies. The tool used was ROB2 and the results can be seen in the supplementary material.

Sterne JAC, Savović J, Page MJ, Elbers RG, Blencowe NS, Boutron I et al. RoB 2: a revised tool for assessing risk of bias in randomised trials. BMJ. 2019; 366: l4898.

  1. I recommend having standardized currency so that we can see a clearer differential in the cost.

Thanks for your interesting contribution. The objective of this review was not to clarify the cost differences, but to show the cost-effectiveness profile of the intervention for future studies.

  1. I suggest the authors add meta-analyses for the cost of the interventions, cost-effectiveness and cost benefits.

We have followed the Joanna Briggs Guide for Systematic Reviews of Economic Evaluations. The guide states that “The real contribution of a systematic review of economic evaluations may not be to produce a single authoritative result, but to help decision makers understand the structure of the resource allocation problem that they are addressing and the impact, on the overall result of the main parameters”.

Thanks to your comment we clarified this and included the guide reference in methods section.

Gomersall JS, MCom BA, Jadotte YT, Xue Y, Lockwood S, Riddle D, et al. Conducting systematic reviews of economic evaluations. International Journal of Evidence-Based Healthcare. 2015(13)3:170-178. doi: 10.1097/XEB.0000000000000063

  1. The statement “Results indicated that nurse-led CM interventions improved all-cause mortality without being statistically significant” (line 313-314) was confusing. Why did the authors state the nurse-led CM interventions improved all-cause mortality but were statistically insignificant, which means no difference between nurse-led CM intervention and the control intervention?

Thanks for your comment, we have modified all the sentences in the abstract, results and discussion sections to give a much more cautious message because the RR estimator was 0.78, 95% but CI was 0.53 to 1.15.

  1. Authors should include the impact of the small size of the study in the limitation.

Thank you for the suggestion, the limitation of small samples is reflected in limitation section. In this sense, “limitations in the systematic review are mostly derived from those of the primary included studies”.

  1. Authors should add recommendations in conclusion.

Thank you for your comment. We have added recommendations in the conclusion for future research. We could not add recommendations for clinical practice as they require the knowledge of other domains (patient values and preferences, stakeholder perspectives, knowledge of the acceptability of the intervention) applying the GRADE methodology.

Round 2

Reviewer 1 Report

What do authors mean by "Results indicated that nurse-led CM interventions may improve all-cause mortality as the findings were only marginally non-significant”. If it is not significant we cannot draw such conclusion. 

Please correct referencing styles- authors use superscripts in the tables and numbers in brackets in the main text 

Author Response

Thank you,

Here we answer reviewer comments:

R: What do authors mean by "Results indicated that nurse-led CM interventions may improve all-cause mortality as the findings were only marginally non-significant”. If it is not significant we cannot draw such conclusion. 

Thank you, we considered your comment and we modified all sentences in reference to mortality.

R:Please correct referencing styles- authors use superscripts in the tables and numbers in brackets in the main text 

Thank you, we corrected all reference styling.

Reviewer 2 Report

MINOR COMMENT:

Dear Authors, 

Thank you for responses. 

1. I2 or I– this should be unified in all txt

Still it is not unified, example: lines 33, 35, 37; Fig 2 , lines 1027, 1040 and other.

2. References are not prepared according to journal’s recommendation ACS Style. 

Author Response

Dear reviewer,

Thank you for your comments.

  1. I2 or I– this should be unified in all txt

We unified in all manuscript.

2. References are not prepared according to journal’s recommendation ACS Style. 

Thank you, we changed all references to ACS Style.